# Synthesis of Ni/Y_2_O_3_ Nanocomposite through USP and Lyophilisation for Possible Use as Coating

**DOI:** 10.3390/ma15082856

**Published:** 2022-04-13

**Authors:** Tilen Švarc, Srećko Stopić, Žiga Jelen, Matej Zadravec, Bernd Friedrich, Rebeka Rudolf

**Affiliations:** 1Faculty of Mechanical Engineering, University of Maribor, Smetanova 17, 2000 Maribor, Slovenia; tilen.svarc@student.um.si (T.Š.); z.jelen@um.si (Ž.J.); matej.zadravec@um.si (M.Z.); 2Process Metallurgy and Metal Recycling, RWTH Aachen University, Intzestrasse 3, 52056 Aachen, Germany; sstopic@ime-aachen.de (S.S.); bfriedrich@ime-aachen.de (B.F.)

**Keywords:** Ultrasonic Spray Pyrolysis (USP), lyophilisation, Ni/Y_2_O_3_ nanocomposite, characterisation, coating

## Abstract

The Ni/Y_2_O_3_ catalyst showed high catalytic activity. Based on this, the aim of this study was to create Ni/Y_2_O_3_ nanocomposites powder with two innovative technologies, Ultrasonic Spray Pyrolysis (USP) and lyophilisation. In the USP process, thermal decomposition of the generated aerosols in an N_2_/H_2_ reduction atmosphere caused a complete decomposition of the nickel (II) nitrate to elemental Ni, which became trapped on the formed Y_2_O_3_ nanoparticles. The Ni/Y_2_O_3_ nanocomposite particles were captured via gas washing in an aqueous solution of polyvinylpyrrolidone (PVP) in collection bottles. PVP was chosen for its ability to stabilise nano-suspensions and as an effective cryoprotectant. Consequently, there was no loss or agglomeration of Ni/Y_2_O_3_ nanocomposite material during the lyophilisation process. The Ni/Y_2_O_3_ nanocomposite powder was analysed using ICP-MS, SEM-EDX, and XPS, which showed the impact of different precursor concentrations on the final Ni/Y_2_O_3_ nanocomposite particle composition. In a final step, highly concentrated Ni/Y_2_O_3_ nanocomposite ink (Ni/Y_2_O_3_ > 0.140 g/mL) and test coatings from this ink were prepared by applying them on a white matte photo paper sheet. The reflection curve of the prepared Ni/Y_2_O_3_ nanocomposite coating showed a local maximum at 440 nm with a value of 39% reflection. Given that Ni is located on the surface of the Ni/Y_2_O_3_ nanocomposite in the elemental state and according to the identified properties, tests of the catalytic properties of this coating will be performed in the future.

## 1. Introduction

Nickel is strategic metal with autocatalytic characteristics, used mostly in organic reactions since many transformations are catalysed by nickel in organometallic chemistry [1,2,3,4,5,6,7]. Experiments show that the Ni matrix in composite coatings is more active than a pure Ni coating [3,4,5,6,7]. Sun et al. [8] have found a crucial role of surface oxygen mobility on nanocrystalline Y_2_O_3_ supports for oxidative steam reforming of ethanol to hydrogen over an Ni/Y_2_O_3_ catalyst. Li et al., have found Ni on a Y_2_O_3_ support to have remarkably efficient catalysis for CO_2_ methanation [9]. Similarly, Taherian et al. [3] showed that yttria-promoted nickel catalysts are as effective in CO_2_ and methane reformation to syngas as more expensive commercial catalysts. Hasan et al. [10] explained the methanation mechanism over Ni/Y_2_O_3_ catalysts successfully. Guo et al. [11] have demonstrated that the catalytic activity is tied intimately to particle size, demonstrating that smaller particles do not necessarily offer better catalytic activity. Based on this, it can be concluded that the particle size and volume-specific surface area are those important properties that have a decisive influence on the functional properties of the materials, especially on nanoparticles. Many methods of nanoparticle synthesis are known, and they are divided into bottom-up and top-down approaches. Most of them are suitable for production of small quantities of nanoparticles with major variations in shapes and sizes of the nanoparticles from the production of different batches. A bottom-up method, called Ultrasonic Spray Pyrolysis (USP), has good potential for removing these technological issues, for a more controlled nanoparticle synthesis [12,13,14,15]. In the first stage, USP uses ultrasound to spray the precursor solutions into droplets, which move with the help of a carrier gas to the reactor, where, in the second stage, the solvent evaporates. In the next stage, the dried droplets with the remaining material are decomposed chemically and reduced at elevated temperatures. In the final stage, nanoparticles are synthesised by the sintering process. The nanoparticles are then transported to the appropriate collection system in the flow of the carrier gas. The aerosol droplet size depends on the chemical-physical characteristics of the used precursor solution and the frequency of the ultrasound [16,17,18]. 

Given that most of the synthesised nanoparticles with USP are usually collected in different suspensions, it is necessary to remove the liquid in order to achieve a powder state of the nanoparticles thus produced by various drying procedures. Freeze drying is offered as one of the options (lyophilisation), where it is possible to obtain nanoparticles in powder form (like a cake), and, thus, achieve their use for many known and new applications [19,20,21,22].

In this scientific discussion, we present the synthesis of the Ni/Y_2_O_3_ nanocomposite through USP synthesis, which was collected in an aqueous solution, and the result was a suspension. The mechanism of Ni/Y_2_O_3_ nanocomposite formation was elucidated with the help of different types of characterisations. The suspensions were further subjected to lyophilisation drying to remove the water, and to give a powder form (like a cake) of the Ni/Y_2_O_3_ nanocomposite. In the next step, ink was prepared from the Ni/Y_2_O_3_ powder, so that test coating lines were made experimentally, on which the colour was measured, and the resulting microstructure was evaluated. The potential possibility of using Ni/Y_2_O_3_ nanocomposites for feasible new applications was investigated in this way.

## 2. Materials and Methods

### 2.1. Materials

Two chemical compounds were used as starting materials for the synthesis of the Ni/Y_2_O_3_ nanocomposite: Nickel nitrate hexahydrate (Ni(NO_3_)_2_ × 6 H_2_O), 98%, hygroscopic, (ThermoFisher, Kandel, Germany) and yttrium oxide hexahydrate (Y(NO_3_)_3_ × 6 H_2_O), 99.8%, (Merck, Darmstadt, Germany).

### 2.2. Synthesis Process

#### 2.2.1. USP Synthesis

Table 1 shows an experimental plan for the preparation of three different precursor solutions, depending on the molar concentrations of each precursor in 1 L of de-ionised (D.I.) water, which were refilled in the ultrasonic generator, PRIZNano (Kragujevac, Serbia) of the USP device (as presented in Figure 1). From the experiments, three types of Ni/Y_2_O_3_ nanocomposites were obtained, labelled as Ni/Y_2_O_3_-0.100/0.200, Ni/Y_2_O_3_-0.050/0.200, and Ni/Y_2_O_3_-0.025/0.100. The USP technological parameters were determined based on our previous implementations of USP syntheses with similar materials [18,23,24]. The precursor solution was subjected to a generator frequency f = 1.7 MHz, resulting in the formation of micro-sized droplets, which were then transported by the use of a carrier gas (N_2_) to the USP reactor. It is composed of two zones: evaporation and reaction. In the first zone (T = 100 °C), droplets were subjected to the evaporation of D.I. water, while, in the second heating zone (T = 900 °C) with the addition of hydrogen (H_2_), the remaining material in the dried droplets was decomposed thermally, reduced, and finally precipitated into Ni/Y_2_O_3_ nanocomposites. The gas flows were as follows: 1 L/min for N_2_ and 1.0 L/min for H_2_.

#### 2.2.2. Lyophilisation

The suspensions with Ni/Y_2_O_3_ nanocomposites were stabilised sterically and cryo stabilised with polyvinylpyrrolidone (PVP) (5 g/L) before the drying process (lyophilisation). PVP was chosen since it is remarkably useful and effective in nanoparticle synthesis, where it can perform functions such as a colloidal stabiliser, a cryo stabiliser, surfactant, shape directing agent, dispersant, and even a reducing agent [19,25,26,27]. The process of drying, or transition from the suspension phase to dry form, is based on the direct application of the nanocomposite suspension and the removal of the solvent by the lyophilisation process. The lyophilisation process is well known for the drying of nanoparticles in pharmaceuticals [28,29]. The drying of the nanosuspension was carried out in several steps to ensure a successful drying process. The freeze-drying process consists of a freezing phase in which the nanosuspension freezes and the solvent is converted into a crystalline or amorphous solid, followed by a rapid pressure drop in the system during the primary and secondary drying phases [30]. It is important that the freeze-drying process is carried out under optimised conditions to prevent destabilisation of the system (e.g., formation of turbid suspensions, sedimentation of nanoparticles, agglomeration) and associated changes in the properties of the nanoparticles.

Lyophilisation was performed using an LIO-2000 FLT lyophiliser, produced by Kambič, Slovenia [31]. The lyophilisation protocol can be seen in Figure 2, from which we can determine that the freezing phase took place during the first 5 h of the process with a shelf temperature of −40 °C. The primary drying phase took place for approximately 6 h at −20 °C, followed by the second phase at −10 °C. After a total of 32 h, the temperature was increased to 10 °C to assess the finalisation of the drying process.

### 2.3. Ink Preparation from the Ni/Y_2_O_3_ Nanocomposite Powder

The lyophilised Ni/Y_2_O_3_ nanocomposite powder Ni/Y_2_O_3_-0.025/0.100 was used to prepare the ink (see Table 1). Ni/Y_2_O_3_-0.025/0.100 was chosen based on the SEM results, which indicated the most uniform distribution of Ni between the particles. The preparation process was performed by dissolving the Ni/Y_2_O_3_ nanocomposite powder in distilled water so that the achieved Ni/Y_2_O_3_ nanocomposite concentration was 0.140 g/mL. Using a brush, we applied the resulting ink to white matte photo paper. A compact coating was formed, which was subjected to further investigations. 

### 2.4. Characterisation

#### 2.4.1. Thermo Gravimetric Analysis (TGA) and Differential Thermal Analysis (DTA) 

In understanding the USP synthesis and preparing Ni/Y_2_O_3_ precursor solutions, TGA analysis, and DTA were done to determine the chemical decomposition reactions of Ni(NO_3_)_2_ × 6 H_2_O and Y(NO_3_)_3_ × 6 H_2_O as starting materials. This was, namely, important, because the decomposition temperature of the chemical components is related closely to setting the required temperature in the reaction zone of the USP device for the synthesis of Ni/Y_2_O_3_ nanomaterials, and to gain insight into the possible formation of intermediate by-products if the conditions were right. The TGA/DTA analysis is, in general, strongly dependent on the kinematic parameters (heating rate, carrier gas, and flow rates) and thermodynamic parameters (decomposition temperature ranges). TGA analyses were made on a TGA 2 device (METTLER TOLEDO, Switzerland) in the temperature range of 50–800 °C, with a heating rate of 10 °C/min and a purge rate of 100 mL/min N_2_. TGA/DTA analyses were performed in an inert atmosphere.

#### 2.4.2. Inductively Coupled Plasma Mass Spectrometry (ICP-MS)

The concentrations of Ni and Y in all three synthesised Ni/Y_2_O_3_ nanocomposite suspensions after USP synthesis were measured with Inductively Coupled Plasma-Mass Spectrometry (ICP-MS). The spectrometer used was an HP, Agilent 7500 CE, equipped with a collision cell (Santa Clara, CA, USA). The following conditions were used for ICP-MS: The power was 1.5 kW, Nebuliser-Meinhard, plasma gas flow was 15 L/min, nebuliser gas flow was 0.85 L/min, make-up gas flow was 0.28 L/min and the reaction gas flow was 4.0 mL/min. The instrument was calibrated with matrix matched calibration solutions. The relative measurement uncertainty was estimated as ±3%. 

#### 2.4.3. Scanning Electron Microscopy with an Energy-Dispersive X-ray Spectroscope (SEM-EDX)

A Scanning Electron Microscope (SEM), Sirion 400 NC (FEI Sirion 400 NC, FEI Technologies Inc., Hillsboro, OR, USA), with an Energy-Dispersive X-ray spectroscope (EDX) INCA 350 (Oxford Instruments, UK), was used for the SEM investigations of the Ni/Y_2_O_3_ nanocomposite powder after lyophilisation and prepared coating on a paper. The Ni/Y_2_O_3_ nanocomposite powder was put on SEM holders with conductive carbon adhesive tape, while the prepared coatings were located directly on the SEM holder with conductive carbon adhesive tape without any additional treatment. The analysis was conducted in a high vacuum with the acceleration voltage of 15–20 kV, using a backscatter detector, at magnifications between 1000 and 50,000×, and by a working distance of 6.8–6.5 mm. By examining the microstructure of the Ni/Y_2_O_3_ nanocomposite powder and based on the SEM micrographs obtained, it was possible to perform nanocomposite particle size and shape analysis and determine their morphology.

EDX was used for the determination of both the qualitative and semi-quantitative chemical composition for the Ni/Y_2_O_3_ nanocomposite particles, and for the upper surface of the prepared coatings. 

#### 2.4.4. X-ray Photoelectron Spectroscopy (XPS)

The X-ray Photoelectron Spectroscopy (XPS or ESCA) analyses were carried out for the Ni/Y_2_O_3_ nanocomposite powder on a PHI-TFA XPS spectrometer (Physical Electronics Inc USA). The spectrometer was equipped with an Al-monochromatic source. The analysed area was 0.4 mm in diameter and the analysed depth was about 3–5 nm. During data processing, the spectra were aligned by setting the C 1s peak at 284.8 eV, characteristic of C-C/C-H bonds. The accuracy of the binding energies was about ±0.6 eV. The XPS method is not sensitive for H and He. Since the sample surface may be contaminated with adsorbed species, we removed a surface layer of thickness of about 10 nm by Ar-ion sputtering, which was applied for 10 min. Since the initial samples Ni/Y_2_O_3_-0.100/0.200, Ni/Y_2_O_3_-0.050/0.200, and Ni/Y_2_O_3_-0.025/0.100 did not have a high enough concentration of Y and Ni to be detected with XPS, consequently, a new concentrated sample of Ni/Y_2_O_3_ nanocomposite powder was prepared from Ni/Y_2_O_3_-0.025/0.100. The procedure was as follows: 5 mL of suspension was stabilised further by sodium citrate (2 g/L) and centrifuged for 30 min at 9000 RPM (Centrifuge Rotina 380R, Andreas Hettich GmbH & Co., Tuttlingen, Germany). After centrifugation 4 mL of supernatant was removed from the sample, while the remaining 1 mL was used for lyophilisation on the basis of the previously presented regime (see Figure 2). The aforementioned step of concentration was used to increase the per weight concentration of Ni, in order to obtain a reliable signal from the XPS detector.

#### 2.4.5. Colour Measurement

The colour of the applied ink coating was analysed using the CIELAB Datacolor SF 600+, Luzern Schweiz, in accordance with the ISO 11664-4 Standard [32].

#### 2.4.6. Statistics

ImageJ software was used for the size measurements of the Ni/Y_2_O_3_ nanocomposite particles from the SEM micrographs. A total of 200 particles were measured in each sample. Bin sizes of 10 nm and 5 nm were used for the size distribution representations. The Mean values and Standard Deviations were calculated from the measured particle sizes for each sample according to the Standard [33].

## 3. Results

### 3.1. TGA/DTA

The TGA/DTA result of (Y(NO_3_)_3_ × 6H_2_O) as starting materials is presented in Figure 3.

Melnikov et al. [34], reported that the thermal decomposition of (Y(NO_3_)_3_ × 6H_2_O) is a complex condensation process, as it creates a tetramer arrangement Y_4_O_4_(NO_3_)_4,_ formed by alternating yttrium and oxygen atoms. The whole condensation process may be described as:4 Y(NO_3_)_3_ × 6H_2_O → 2Y_2_O_3_ + 6N_2_O_5_ + 24H_2_O(1)

According to our TGA/DTA results, thermal decomposition takes place through stepwise reactions as follows:Y(NO_3_)_3_ × 6H_2_O → Y(NO_3_)_3_ + 6H_2_O,(2)
2Y(NO_3_)_3_ → Y_2_O_3_ + 3N_2_O_5_,(3)

As revealed in Figure 3, mass loss starts at early temperatures due to the loss of chemically bound water (the two peaks around 50 and 80 °C). The mass loss related to the peak at 240 °C is below 30%, which corresponds to the stoichiometric loss of 5 water molecules (28.19%), according to the reaction (2). The next peaks at 370 °C can be associated with the joint loss of the remaining 6th crystalline water molecule, which is presenting with the chemical reaction (2). The thermal decomposition of yttrium nitrate to yttrium oxide starts at 400 °C. The loss of the remaining nitrogen, formation of N_2_O_5,_ and generation of yttrium oxide through reaction (3) is represented by the two well-separated peaks at 460 and 500 °C, with an overall mass loss of 40% that is negligibly different from the stoichiometric 40.9%. Hence, the complete transformation to yttrium oxide takes place at around 550 °C. The theoretically calculated mass loss of 70% is in accordance with the 72.01%, determined experimentally using TGA analysis.

The TGA/DTA result of Ni(NO_3_)_2_ × 6H_2_O as starting materials is presented in Figure 4.

As reported by Brockner et al. [35], the thermal decomposition of Ni(NO_3_)_2_ × 6H_2_O, follows a stepwise decomposition, with the first two steps resulting in the removal of 2 water molecules (80 °C), followed by the removal of the remaining 4 crystalline water molecules (150 °C). Partial decomposition of the nitrate begins at 145 °C, following the reactions in Equations (4) and (5). The final decomposition of Ni(NO_3_)_2_ to ‘‘NiO’’ begins at approximately 250 °C, and follows the reaction pattern as shown in Equation (6):Ni(NO_3_)_2_ × 2H_2_O → Ni(NO_3_)(OH) × 2H_2_O + NO_2_,(4)
Ni(NO_3_)(OH)_2_·× H_2_O → Ni(NO_3_)(OH)_1.5_ O_0.25_·H_2_O + 0.25 H_2_O,(5)
Ni(NO_3_)(OH)_1.5_ O_0.25_ × H_2_O → 0.5 Ni_2_O_3_ + HNO_3_ + 1.25 H_2_O,(6)

In order to facilitate the reduction with shorter residence times at the higher temperature of 500 °C with respect to TGA, but simultaneously with the defined structure and size of Y_2_O_3_ particles, H_2_ was utilised in the USP synthesis between 700 °C and 900 °C. 

### 3.2. ICP-MS 

The results of ICP-MS analyses can be seen in Table 2.

The highest concentrations of Ni and Y in the suspensions were measured in Ni/Y_2_O_3_-0.050/0.200, although the baseline concentrations in the precursor were not the highest here, but in Ni/Y_2_O_3_-0.100/0.200. Such results can be attributed to the conditions during USP synthesis, as observed during the Ni/Y_2_O_3_-0.100/0.200 experiment. The Ni/Y_2_O_3_ nanocomposite particles were deposited on the transport tube, resulting in losses, and, thus, a decrease in the concentration of Y and Ni. The lowest concentrations of both elements were measured in the suspension of sample Ni/Y_2_O_3_-0.025/0.100, which is consistent with the concentration in the precursor. 

### 3.3. SEM-EDX

The real particle size is very important for the catalytic application [36,37,38] since particle size is the property that links the surface area to the volume ratio, which varies significantly when particles are on the nanoscale. Guo et al. [11] demonstrated that smaller particles do not necessarily offer better catalytic activity. For this purpose, direct measurement of the Ni/Y_2_O_3_ nanocomposite particle size from SEM micrographs was performed as an additional method. The results of the measurements are shown in Figure 5. In Samples 1 and 2, where the concentrations of Y and Ni in the precursor were higher, more large particles (with a size range >1800 nm) were observed than in Ni/Y_2_O_3_-0.025/0.100, where both the concentrations in the precursors were lower. In addition, most particles (more than 70%) of sample Ni/Y_2_O_3_-0.025/0.100 were with a size below 900 nm, which indicates how to control the synthesis of USP with the aim to achieve the targeted size of homogeneous Ni/Y_2_O_3_ nanocomposite particles.

SEM micrographs 8a–8c show that the Ni/Y_2_O_3_ nanocomposite particles are very spherical in all three samples. A closer examination showed that the presence of two types of particles was detected in Samples 1 and 2, namely the larger ones, which appear to be dark, and the smaller ones, which are bright. EDX analysis found that darker particles are rich in Y and O, while Ni is not at all. The presence of Ni, Y, and O was confirmed for the smaller, lighter particles. In Ni/Y_2_O_3_-0.025/0.100, EDX analysis found that all particles contained Ni, Y, and O and that there was actually no presence of Y-rich particles, see Figure 6.

Sample Ni/Y_2_O_3_-0.025/0.100 was used to prepare the ink, and, later, the coating, due to the balanced/uniform chemical composition of the Ni/Y_2_O_3_ nanocomposite particles with a content of all three elements (Ni, Y, O). SEM/EDX analyses were performed identically on the powder before ink preparation and after the applied coating. The results are shown in Figure 7. The EDX analysis did not detect Ni on the bright particles on the coating, which may be due to the low sensitivity of the detector, or that the ink application process on the paper resulted in a decrease in Ni concentration on the Ni/Y_2_O_3_ nanocomposite particles. The presence of C origin in the applied coating originated from the paper (coating holder), and could not be removed from the EDX analysis.

### 3.4. XPS Analysis

Figure 8 shows the XPS spectrum obtained on the surface of the Ni/Y_2_O_3_ nanocomposite powder of the concentrated sample Ni/Y_2_O_3_-0.025/0.100. The presence of the elements C, O, N and Y were detected via the peaks C 1s, O 1s, N 1s, and Y 3d. The surface chemical composition (surface layer of thickness 3 nm) was 77 at.% of C, 11 at.% of N, 12 at.% of O, and 0.1 at.%. No Ni was detected at the surface. Figure 9 shows the XPS spectrum obtained on the Ni/Y_2_O_3_ nanocomposite powder after removing about 10 nm of the surface layer by Ar-ion sputtering. The presence of the elements C, O, N, Y, and Ni was detected via peaks C 1s, O 1s, N 1s, Y 3d, and Ni 2p. The surface chemical composition after removing the 10 nm thick surface layer was 70 at.% of C, 4 at.% of N, 14 at.% of O, 10 at.% of Y, and 1.1 at.% of Ni. To obtain insight into the surface chemistry, the high-energy resolution XPS spectra were acquired to obtain insight into the surface chemistry. Figure 10 shows the Y 3d spectrum obtained after removing the 10-nm thick layer. It consists of the Y 3d5/2 peak at 157.8 eV and the Y 3d3/2 peak at 159.8 eV. The binding energy of the Y 3d5/2 peak at 157.8 eV is often related to the Y_2_O_3_ compound, where a Y(3+) oxidation state is present. Figure 11 shows the Ni 2p3/2 spectrum obtained after removing the 10-nm thick layer. It consists of the main peak at 853.0 eV and a small satellite peak at 859.9 eV. The binding energy of the main peak 853.0 eV is probably related to the Ni(0) metallic state [39,40].

### 3.5. Colour of the Ni/Y_2_O_3_ Nanocomposite Coating

Figure 12 shows the reflection curve of the prepared Ni/Y_2_O_3_ nanocomposite coating. A local maximum can be seen at 440 nm with a value of 39% reflection (as shown with the dashed vertical line). With the further increase in the wavelength, a small decline in reflection can be seen up to 470 nm. Using an additional increase in the wavelength a slow increase in reflection can be observed, reaching the maximum value of 44% at 700 nm. The drop in reflectance at 400 nm, suggests a higher absorption in the blue/UV wavelengths. 

The chromaticity diagram was prepared based on the measured colour of the Ni/Y_2_O_3_ nanocomposite coating and the results as shown in Table 3—see Figure 13. After inserting the coordinates of the CIELAB colour space in the chromaticity diagram, the colour of the Ni/Y_2_O_3_ nanocomposite coating can be presented as a light grey-silver colour.

## 4. Discussion

The performed tests confirmed the hypothesis that successful synthesis of the Ni/Y_2_O_3_ nanocomposite with USP and lyophilisation is possible. If we analyse the USP synthesis in more detail, the next finding is that an aqueous solution of nickel and yttrium nitrate is appropriate, as it allows the formation of relatively chemically homogeneous and size-related nanoparticles at an optimal molar ratio of Ni/Y nitrate (0.25). In the reactor part of the USP device, water evaporation takes place first, and then the dried droplets enter the high-temperature area (T = 900 °C). The first step in this is the thermal decomposition of yttrium nitrate and nickel nitrate to yttrium oxide and nickel oxide. Because of the high stability of yttrium oxide in comparison to nickel oxide, the hydrogen reaction is possible only for nickel formation. Therefore, the formation of nickel/yttrium oxide is possible after dehydration and thermal decomposition of the metal nitrates, and hydrogen reduction of nickel oxide is only possible in an H_2_/N_2_ atmosphere [35]. The reduction of Ni_2_O_3_ to elemental Ni is first mediated by the decomposition to NiO, as shown in Equations (7) and (8).
3Ni_2_O_3_ → 2Ni_3_O_4_ + 0.5O_2_,(7)
Ni_3_O_4_ → 3NiO + 0.5O_2_,(8)

The thus formed NiO is reduced by H_2_ to elemental Ni, following the reaction shown in Equation (9).
3NiO + 3/2H_2_ → 3Ni + 3/2H_2_O,(9)

It is assumed that the residence time (about 3 s for the system in Figure 1) is too long for the formation of a uniform phase of the Ni-doped Y_2_O_3_ matrix. This is due mainly to the faster formation of the Y_2_O_3_ phase, as is evident in its significantly lower standard enthalpy of formation of −1905.31 kJ/mol [41]. While the formation enthalpies of Ni_2_O_3_ (−244.3 kJ/mol [42])_,_ NiO (−489.5 kJ/mol [43]), and Ni (430.12 kJ/mol [44]), with the enthalpy for Ni_3_O_4_ being unknown, this is an unstable intermediate compound. With most considered enthalpies the combined formation enthalpy of Ni indicates a much slower decomposition to elemental Ni. 

The hypothesised mechanism of synthesis follows the initial formation of pure Y_2_O_3_, with no dissolved Ni_2_O_3_. As the Y_2_O_3_ particle grows it produces a surface crust that is rich in NiO. This crust can be ejected to form separate smaller particles, or it can consolidate on the surface. In both cases, the following reduction of NiO to elemental Ni is mediated by hydrogen, thus forming an Ni-doped Y_2_O_3_ matrix (as presented in Figure 14). Experimental studies of the obtained powders brought us to the conclusion that the mechanism of nickel/yttrium oxide preparation from mixed nickel nitrate and yttrium oxide is a complex one, and that it proceeds gradually from dehydration, thermal decomposition of the metal nitrate, and hydrogen reduction of nickel oxide to nickel.

The hypothetical short residence time of about 3 s is not enough to complete all the needed chemical reactions. On the other hand, the particle size depends firstly on the droplet size and transport phenomena, which enable the coalescence and coagulation of droplets. The particle size is increased as a consequence of this behaviour. An increase in the concentration at the same temperature leads to a decrease in particle size. 

Drying Ni/Y_2_O_3_ nanocomposite suspensions by freeze-drying and monitoring sample temperatures at the bottom of the sample cake shows that the drying cycle could be shortened by at least 10 h (Figure 2). In a further process development, the drying time by freeze-drying depends strongly, not only on the temperatures being selected properly for the particular material but also on the sample height. The temperatures and pressures used in the present study were chosen to control the drying process conservatively. The goal was not to lose the nanocomposite material during severe drying, resulting in high velocities of the sublimed solvent toward the condenser where de-sublimation of the solvent occurs. If the drying temperatures are increased and the pressure [45] is decreased in the primary and secondary drying phases, the process will lead to a shorter drying phase, but the desired functionalities of the nanocomposite might be lost.

In summary, the XPS results show the presence of Y(3+) and Ni(0) oxidation states. This may indicate that mainly a Y-oxide phase is present on the Y-Ni particles. We should note that Ar-ion sputtering, which we applied for the removal of the surface layer, may change (reduce) the oxidation states of elements due to the surface damage introduced by the ion bombardment. In this sense, the presence of the Ni-oxide also cannot be excluded.

## 5. Conclusions

The performed study describes the synthesis of an Ni/Y_2_O_3_ nanocomposite powder that was made by the USP and lyophilisation methods, and this approach cannot be found in the scientific literature. Furthermore, the successful preparation of the ink and test coating is demonstrated by the highest concentration of Ni/Y_2_O_3_. 

The following scientific conclusions can be drawn:-An Ni/Y nitrate-based aqueous solution as a precursor allows USP synthesis of Ni/Y_2_O_3_ nanocomposite particles, which are collected in an aqueous PVP suspension.-Lyophilisation proved to be a suitable process for water removal and for obtaining nanocomposite Ni/Y_2_O_3_ powder.-ICP-MS and SEM/EDX analyses of the Ni/Y_2_O_3_ nanocomposite powder showed the impact of precursor concentrations on the final particle formation and their composition.-XPS research confirmed that the Ni/Y_2_O_3_ nanocomposite particles are composed of elemental Ni^0^ and Y_2_O_3_.-The mechanism of Ni/Y_2_O_3_ nanocomposite formation was set up with the initial formation of pure Y_2_O_3_ and Ni doping on its surface.-The prepared Ni/Y_2_O_3_ nanocomposite ink allowed the preparation of a coating that has a light grey-silver colour.

In the future, Ni/Y_2_O_3_ coatings can be analysed further to assess their use as a catalyst.

## Figures and Tables

**Figure 1 materials-15-02856-f001:**
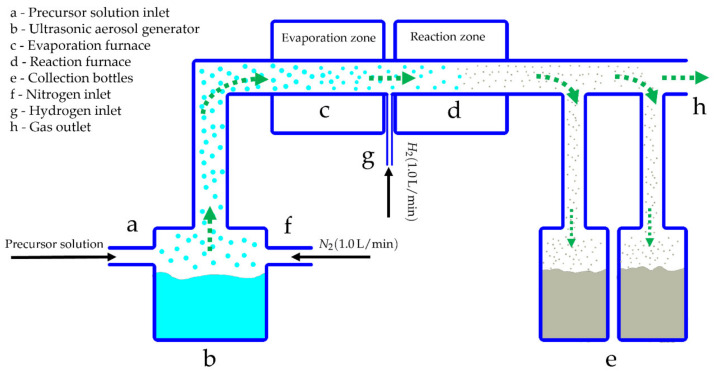
Schematic presentation of the USP device.

**Figure 2 materials-15-02856-f002:**
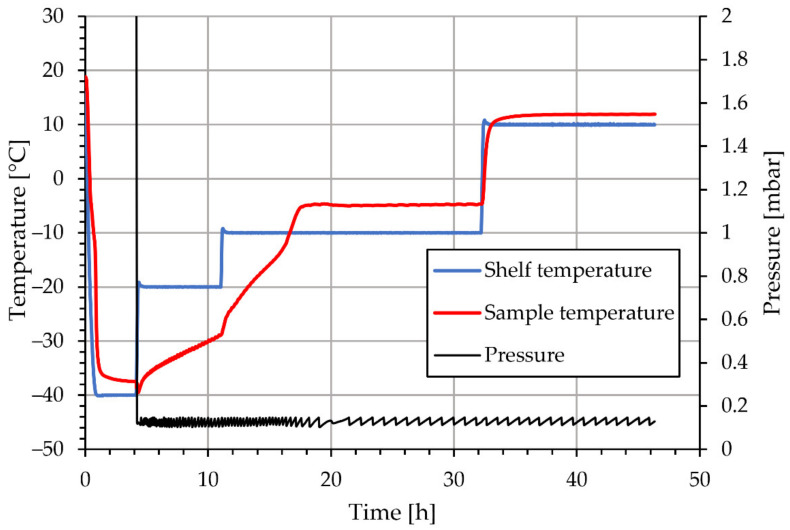
Lyophilisation protocol, shelf temperature, sample temperature and the pressure inside the lyophilisatior for Ni/Y_2_O_3_ nanocomposite suspensions.

**Figure 3 materials-15-02856-f003:**
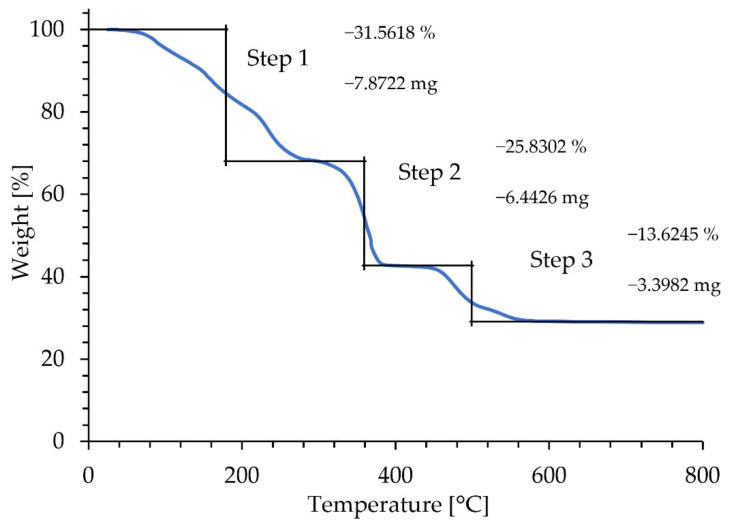
TGA plot of the thermal decomposition of Y(NO_3_)_3_ × 6 H_2_O.

**Figure 4 materials-15-02856-f004:**
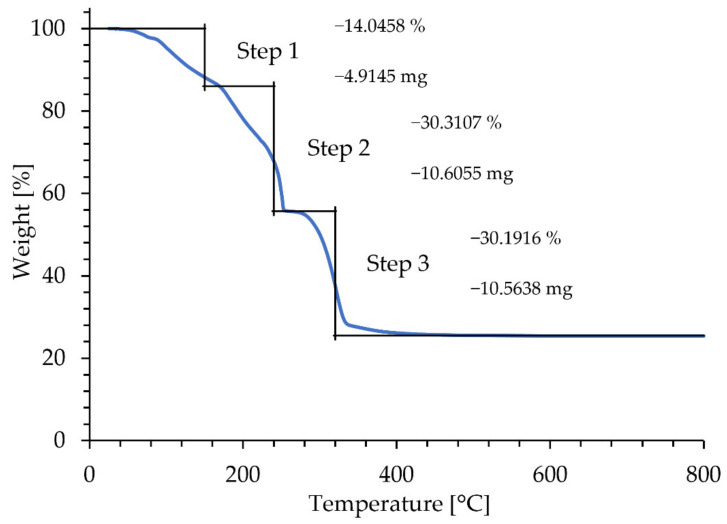
TGA plot of the thermal decomposition of Ni(NO_3_)_2_ × 6 H_2_O.

**Figure 5 materials-15-02856-f005:**
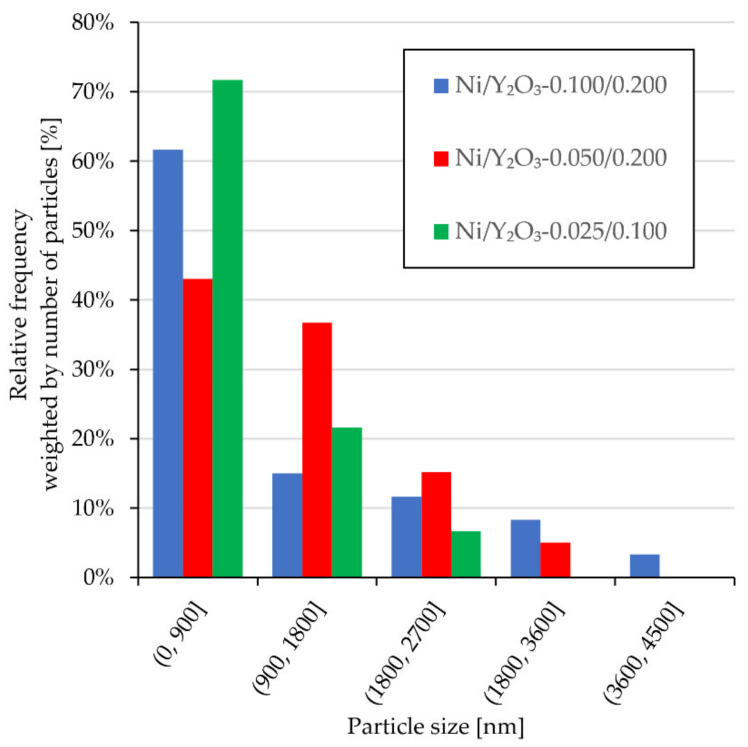
Ni/Y_2_O_3_ nanocomposite particle distribution obtained from the SEM micrographs for samples: Ni/Y_2_O_3_-0.100/0.200, Ni/Y_2_O_3_-0.050/0.200, Ni/Y_2_O_3_-0.025/0.100.

**Figure 6 materials-15-02856-f006:**
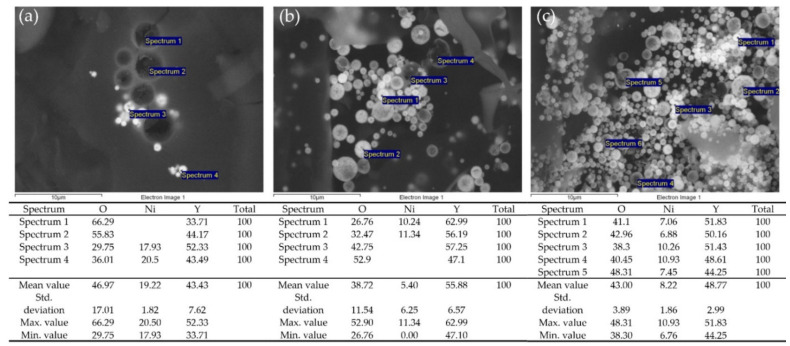
SEM micrographs and EDX analysis of the powder: (**a**) Ni/Y_2_O_3_-0.100/0.200, (**b**) Ni/Y_2_O_3_-0.050/0.200, and (**c**) Ni/Y_2_O_3_-0.025/0.100.

**Figure 7 materials-15-02856-f007:**
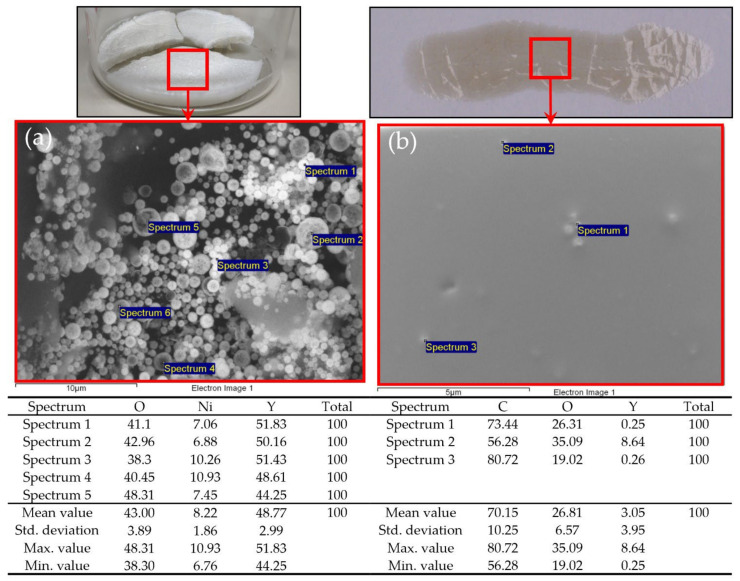
SEM/EDX analysis of the: (**a**) dried Ni/Y_2_O_3_-0.025/0.100 sample and (**b**) applied coating.

**Figure 8 materials-15-02856-f008:**
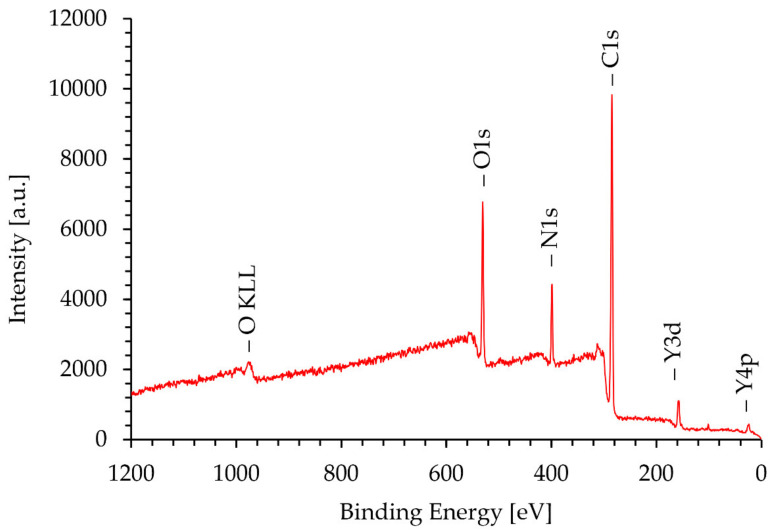
XPS spectrum of the concentrated sample Ni/Y_2_O_3_-0.025/0.100 surface.

**Figure 9 materials-15-02856-f009:**
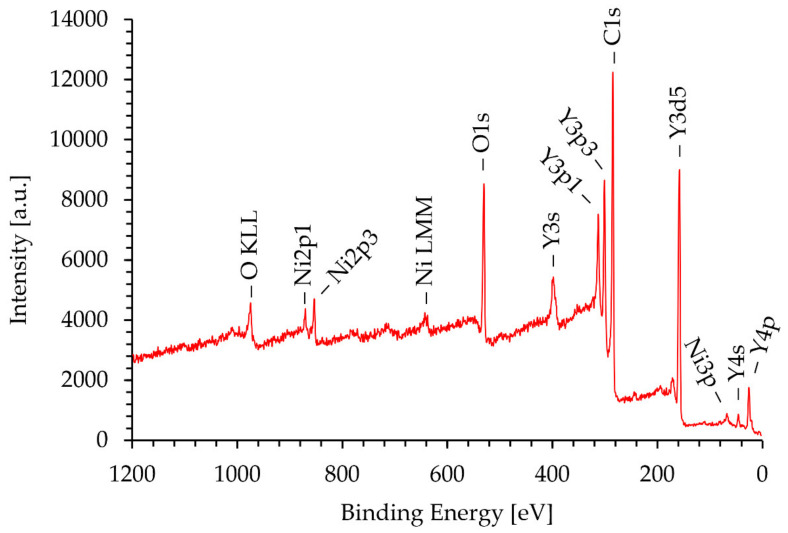
XPS spectrum of the concentrated sample Ni/Y_2_O_3_-0.025/0.100 after removal of a 10 nm thick surface layer.

**Figure 10 materials-15-02856-f010:**
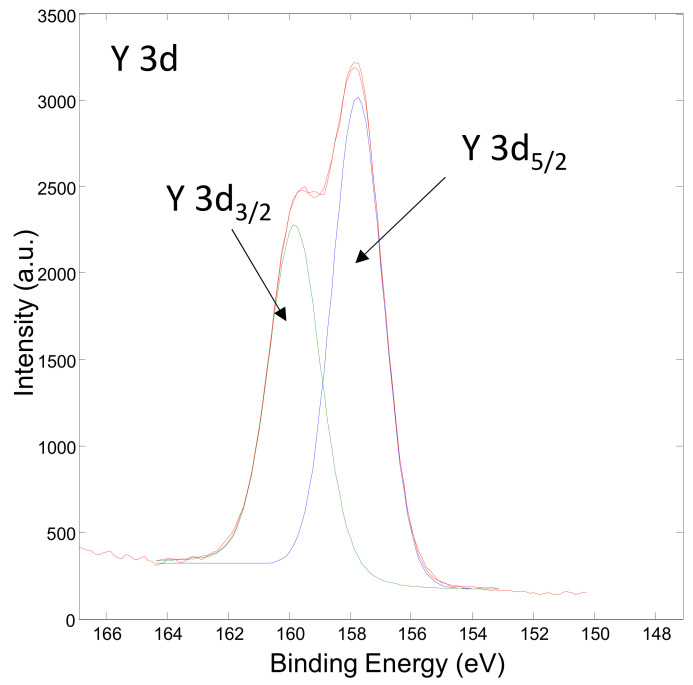
XPS spectrum of Y 3d obtained after removing the 10-nm thick layer on the concentrated sample Ni/Y_2_O_3_-0.025/0.100.

**Figure 11 materials-15-02856-f011:**
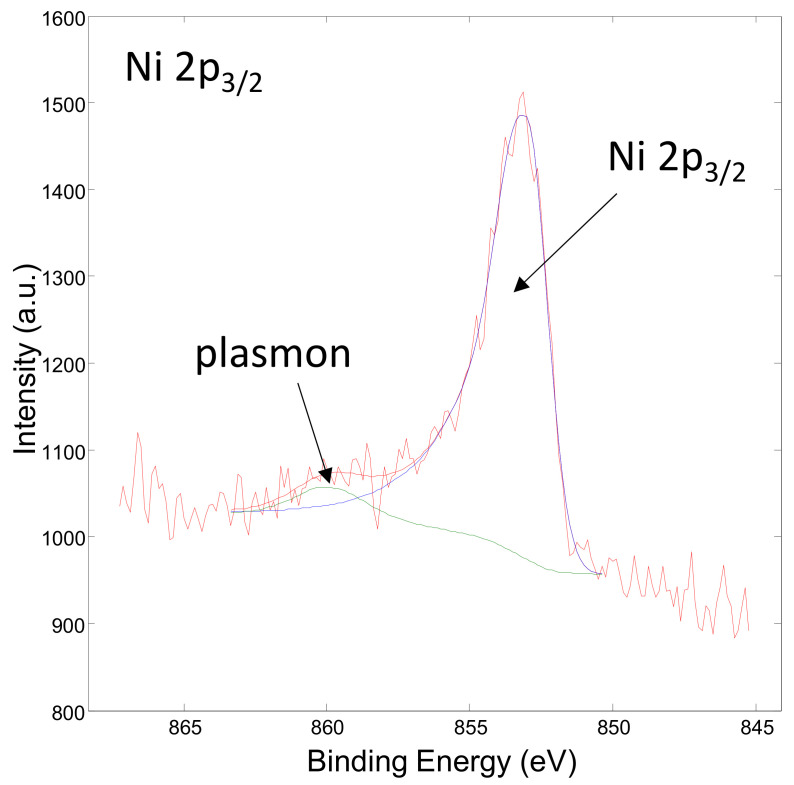
XPS spectrum of Ni 2p obtained after removing the 10-nm thick layer on the concentrated sample Ni/Y_2_O_3_-0.025/0.100.

**Figure 12 materials-15-02856-f012:**
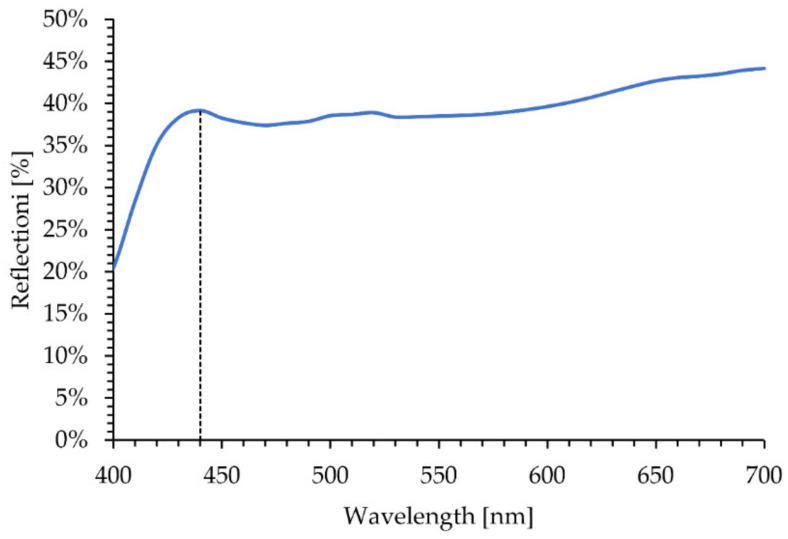
Reflection curve of Ni/Y_2_O_3_ nanocomposite coating, obtained with CIELAB analysis.

**Figure 13 materials-15-02856-f013:**
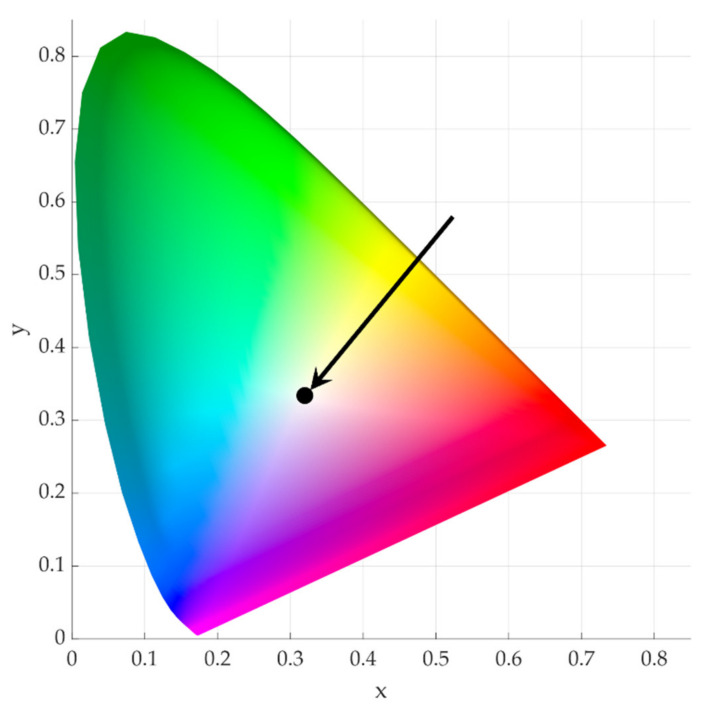
Chromaticity diagram, showing the measured colour of Ni/Y_2_O_3_ nanocomposite coating.

**Figure 14 materials-15-02856-f014:**
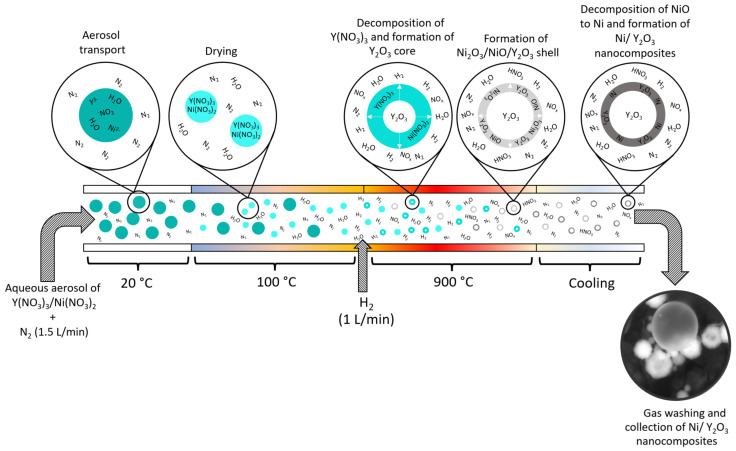
Schematic presentation of Ni/Y_2_O_3_ nanocomposite particles through USP.

**Table 1 materials-15-02856-t001:** Experimental plan for preparation of the Ni/Y_2_O_3_ precursor solutions.

	Concentration [mol/L]		Ni/Y Nitrate
	Ni(NO_3_)_2_ × 6 H_2_O	Y(NO_3_)_3_ × 6 H_2_O	
Ni/Y_2_O_3_-0.100/0.200	0.100	0.200	0.50
Ni/Y_2_O_3_-0.050/0.200	0.050	0.200	0.25
Ni/Y_2_O_3_-0.025/0.100	0.025	0.100	0.25

**Table 2 materials-15-02856-t002:** The concentration of nickel and yttrium in the USP synthesised suspensions.

Suspension Ni/Y_2_O_3_	Ni [μg/mL]	Y [μg/mL]
Ni/Y_2_O_3_-0.100/0.200	87.3	263.0
Ni/Y_2_O_3_-0.050/0.200	100.3	568.8
Ni/Y_2_O_3_-0.025/0.100	29.3	185.8

**Table 3 materials-15-02856-t003:** CIELAB colour space coordinates of the Ni/Y_2_O_3_ nanocomposite coating obtained with the CIELAB analysis.

L*	a*	b*	C*	h
77.134	1.398	1.846	2.322	52.434

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
