# Peer review of "Synthesis of Ni/Y2O3 Nanocomposite through USP and Lyophilisation for Possible Use as Coating"

_materials, 2022, doi:10.3390/ma15082856_

Round 1

Reviewer 1 Report

The authors have described the synthesis of Ni/Y2O3 nanocomposite considering Ultrasonic Spray Pyrolysis and lyophilisation methods. These nanocomposites have been synthesized in order to be used as coating. The nanocomposites have been characterized by DTA, DLS, EDX, ICP-MS, SEM, TGA and XPS. Considering a suited concentration of Ni, the nanocomposite can be used as coating and clearly show that it had the light grey - silver colour.

I reject this article and I think that the quality of this article has to be improved before another submission.

1) The authors should give more references in order to illustrate their hypotheses.

2) The authors should reconsider the naming of the samples in order to have a better idea of the differences between the different samples.

3) The authors should improve the quality of the characterization part. For example, for DLS analyses, the authors should give the concentration of the samples for the DLS analyses. Have these analyses been triplicated ? The authors should give the size distributions and the autocorrelation functions and not only the Z average diameter values.

4) For the zeta potential values, the authors should give the zeta potential distributions and the phase plot figures.

5) For the size distribution obtained by SEM, the authors should give one size distribution for one sample in one figure and not the size distributions of all the samples in only one figure.

6) Why did the authors only realize the TgA analysis of the nickel and yttrium salts ?

7) The authors should explain why they used sample 3 as coating and not sample 1 or sample 2.

8) The authors should explain why they used sample 4 for XPS analysis.

Author Response

Review 1 We thank the Reviewer for the constructive and insightful comments. 

The authors have described the synthesis of Ni/Y2O3 nanocomposite considering Ultrasonic Spray Pyrolysis and lyophilisation methods. These nanocomposites have been synthesized in order to be used as coating. The nanocomposites have been characterized by DTA, DLS, EDX, ICP-MS, SEM, TGA and XPS. Considering a suited concentration of Ni, the nanocomposite can be used as coating and clearly show that it had the light grey - silver colour.

I reject this article and I think that the quality of this article has to be improved before another submission.

  • The authors should give more references in order to illustrate their hypotheses.

 Reply:  More references from the years 2015, 2016, 2017, 2018, 2019 and 2020 have been added in the Introduction as the reviewer recommended.

  1. Taherian, Z.; Khataee, A.; Orooji, Y. Facile synthesis of yttria-promoted nickel catalysts supported on MgO-MCM-41 for syngas production from greenhouse gases. Renew. Sustain. Energy Rev. 2020, 134, 110130. 10.1016/j.rser.2020.110130.
  2. Abdullah, B.; Abd Ghani, N.A.; Vo, D.V.N. Recent advances in dry reforming of methane over Ni-based catalysts. J. Clean. Prod. 2017, 162, 170–185. 10.1016/j.jclepro.2017.05.176.
  3. Aramouni, N.A.K.; Touma, J.G.; Tarboush, B.A.; Zeaiter, J.; Ahmad, M.N. Catalyst design for dry reforming of methane: Analysis review. Renew. Sustain. Energy Rev. 2018, 82, 2570–2585. 10.1016/j.rser.2017.09.076.
  4. Abdulrasheed, A.; Jalil, A.A.; Gambo, Y.; Ibrahim, M.; Hambali, H.U.; Shahul Hamid, M.Y. A review on catalyst development for dry reforming of methane to syngas: Recent advances. Renew. Sustain. Energy Rev. 2019, 108, 175–193. 10.1016/j.rser.2019.03.054.
  5. Park, J.H.; Yeo, S.; Chang, T.S. Effect of supports on the performance of Co-based catalysts in methane dry reforming. J. CO2 Util. 2018, 26, 465–475. 10.1016/j.jcou.2018.06.002.
  6. Koczkur, K.M.; Mourdikoudis, S.; Polavarapu, L.; Skrabalak, S.E. Polyvinylpyrrolidone (PVP) in nanoparticle synthesis. Dalt. Trans. 2015, 44, 17883–17905. 10.1039/c5dt02964c.
  7. Bhattacharjee, S. DLS and zeta potential - What they are and what they are not? J. Control. Release 2016, 235, 337–351. 10.1016/j.jconrel.2016.06.017.
  • The authors should reconsider the naming of the samples in order to have a better idea of the differences between the different samples.

Reply: The naming of the samples was reconsidered. The sample name now includes the precursor molar concentrations. The new sample names follow the following formula, Ni/Y2O3-[Nickel precursor concentration]/ [Yttrium precursor concentration].

  • Sample 1 → Ni/Y2O3-0.100/0.200
  • Sample 2 → Ni/Y2O3-0.050/0.200
  • Sample 3 → Ni/Y2O3-0.025/0.100
  • The authors should improve the quality of the characterization part. For example, for DLS analyses, the authors should give the concentration of the samples for the DLS analyses. Have these analyses been triplicated? The authors should give the size distributions and the autocorrelation functions and not only the Z average diameter values.

Reply: The concentration of the suspensions was measured with ICP-OES as is shown in Table 3.

The USP syntheses were conducted as described in section 2.2. and were not repeated. The DLS analyses were conducted with one sample for each of the samples 1-3. However, the DLS machine and its corresponding software determine the optimal number of repetitions for one sample automatically. The data obtained from the DLS analysis were not evaluated additionally statistically, as they were already evaluated by the software of the DLS machine.

The size distribution plot was added as Figure 5. The autocorrelation functions were not added, as the main focus of the article is not the DLS analysis. However, we will add them in the Supplement 1 to the article – Part Supplementary material.

  • For the zeta potential values, the authors should give the zeta potential distributions and the phase plot figures.

Reply: The zeta potential distribution was added as Figure 6. Similarly as the autocorrelation function, the phase plots were not added in the article, and were rather added in the Part Supplementary material (Supplement 2).

  • For the size distribution obtained by SEM, the authors should give one size distribution for one sample in one figure and not the size distributions of all the samples in only one figure.

Reply: This was a point of debate between the authors prior to sending in the article. There are two main reasons we have decided to leave the size distribution of all samples in one Figure.

  1. The article already has 16 Figures and with the addition of more Figures it would become cluttered.
  2. We believe that the comparison between the numbers of particles in each interval can be conducted more clearly when all the distributions are on the same Figure.
  • Why did the authors only realize the TgA analysis of the nickel and yttrium salts?

Reply: The TgA analysis was conducted with the goal to determine the chemical and physical characteristics of the precursors, as described in part 2.5.1 lines 136-148. The TgA analysis provided information on the required temperature in the USP reaction zone to allow Ni/Y2O3 nanocomposite synthesis to take place. If the temperature was too low in the USP process the reaction would not take place, and the Ni/Y2O3 nanocomposite would not be sintered. If the temperature was too high an explosive decomposition would occur (the GTP (gas-to-particles) mechanism would predominate), and this would result in uneven damaged nanocomposite particles that do not have the desired shape, morphology, etc.

  • The authors should explain why they used sample 3 as coating and not sample 1 or sample 2.

Reply: The sample 3 – now Ni/Y2O3-0.025/0.100, was chosen for coating based on the SEM/EDX results, which indicated the most uniform distribution of Ni. A clearer explanation was added in lines 129 and 130.

  • The authors should explain why they used sample 4 for XPS analysis.

Reply: XPS analysis on samples 1-3 did not show any signal for nickel or yttrium, as the concentration of the particles was too low, and, for that reason, sample 4 was prepared. Sample 3 (Ni/Y2O3-0.025/0.1) was chosen for the same reason as in the coating preparation (point 7) in this review. A clearer explanation was added in lines 213, 219 and 220.

Reviewer 2 Report

Comments in the attached file.

Author Response

Review 2 We thank the Reviewer for the constructive and insightful comments.

At first – some small typos are present in the text – double space or comma. Furthermore, the “Table 3” name is present in the text for two different tables – needs to be corrected.

These issues have been corrected.

For scientific point of view, measured Zeta potential lower than 20mV indicates that suspension is not stable at all, however it is possible that stabilization mechanism is different than electrostatic repulsion forces, maybe steric stabilization occurs.

The reviewer is correct, the suspension is not stable on its own. For that reason, PVP was added, since it acts as both a steric stabiliser for the liquid suspension and a cryo stabiliser (cryo protectant) during freezing and freeze drying.

Than it should be explained in the text how it works.

A short explanation has been added with an additional reference in lines 102-106 and 172-176.

It is possible, that PAA residuals are present in the surface of particles and act as steric stabilizer, but still the Zeta potential should be higher in this case. Maybe it is worth to measure Zeta potential in lower concentration and in the function of pH to find the “Zero charge” pH.

PAA was not used in the presented article. If PAA refers to peracetic acid, we must conclude that, even if there are any similar residuals caused by the thermal decomposition of PVP in the collection solution, that the main stabilisation mechanism of steric stabilisation by PVP is the dominant factor in the described system.

Measuring the zeta potential at a lower concentration and as a function of pH to find a “zero charge” pH is not necessary, because primary stabilisation takes place via spherical stabilisation, as PVP stabilisation is independent of pH.

Round 2

Reviewer 1 Report

I would like to thank the authors for this revised version.

Concerning the modifications that have been realized, I suggest the authors not to talk about the DLS results and zeta potential results because the autocorrelation functions are not linked to usable data and this is the same thing for zeta potential analyses. If the authors decide to have these analyses in their article, the experimental conditions have to be improved in order to have usable analyses.

Author Response

Reply:

Based on your comment regarding the DLS analysis, we have decided to omit the sections containing the DLS and zeta potential analysis, since we can no longer carry out additional analysis to improve the experimental conditions. As the DLS analysis is not an integral part to the paper, we believe it is still suitable for publication. We thank you for your review.
